# Comparison of Fire Behaviors of Thermally Thin and Thick Rubber Latex Foam under Bottom Ventilation

**DOI:** 10.3390/polym11010088

**Published:** 2019-01-08

**Authors:** Qi Yuan, Dongmei Huang, Yiwei Hu, Liming Shen, Long Shi, Mingzhen Zhang

**Affiliations:** 1College of Quality and Safety Engineering, China Jiliang University, Hangzhou 310018, China; solar9552@163.com (Q.Y.); a595713616@163.com (Y.H.); zhangmingzhen2679@163.com (M.Z.); 2Xilinmen Furniture Co., Ltd., Shaoxing 312000, China; slm624167421@126.com; 3Civil and Infrastructure Engineering Discipline, School of Engineering, RMIT University, Melbourne 3001, Australia; long.shi@rmit.edu.au

**Keywords:** rubber latex foam, fire behavior, thermally thin materials, thermally thick materials

## Abstract

Fire behaviors of rubber latex foam under different thickness conditions (*d* = 1, 2, and 5 cm) were explored by using a self-built small-scale experimental platform. It can be shown that the flame spread menchanism of thermally thin and thermally thick rubber latex foam is different. Rubber latex foam with a thickness of 2 cm shows higher fire risk, whose value of flame spread rate, maximum flame height, maximum mass loss rate, and maximum temperature are 2.93 × 10^−3^ m/s, 851.88 mm, and 1.83 g/s, 948.00 °C, respectively. On the one hand, this may due to the different mechanisms of flame spread, resulting in different preheating zones on the surface. On the other hand, this may because the thickness of residue formed by thermally thick materials is larger than the thin ones, obstructing the contact of the rubber latex foam with fresh air. In addition, a special phenomenon is noticed during the stage II, where the bottom unburned zone is located in the four edges (thermally thin material) and middle player (thermally thick material).

## 1. Introduction

Nowadays, rubber latex foam has been widely used as filler for high-end mattresses on account of its strong resilience, stability, good breathability, anti-mite sterilization, promotion of sleep, and so on [1,2], as seen in Figure 1. The microstructure of rubber latex foam is a three-dimensional porous structure [3,4,5]. In addition, this material is evenly distributed, with some through-holes to improve the permeability of rubber latex foam and degree of comfort when people use it, as seen in Figure 1b. However, in the event of a fire, the cellular structure and the through-holes greatly accelerate the spread of fire, and increase the full development of the combustion due to a high specific surface and well ventilation, causing great casualties and huge property losses [6].

At present, considering the high price of rubber latex foam (as shown in Figure 1), it is usually not used directly, but as the upper filling layer of the mattress. As shown in Figure 2, common mattresses are divided into five layers from top to bedplate, namely fabric layer, rubber latex foam layer, other cushion filling layer, irony spring layer, and other cushion filling layer. Cotton, linen, and silk materials are usually used as surface coatings. Other common cushion layers include polyurethane foam, cotton, etc. It should be emphasized that there is a connecting layer between the rubber latex foam and the iron spring layer, named other cushion filling layer, usually with good permeability. The permeability of the iron spring layer and the third layer of different mattresses is different, which makes the bottom of the rubber latex foam in different ventilation conditions. Therefore, the study of the fire behavior of rubber latex foam and assessment of the fire risk should take into account some crucial factors, such as the type of surface material, the thickness of rubber latex foam, the number of latex layers, ventilated conditions, and so on.

Although rubber latex foam has been widely used, the recent research only focuses on synthesis technology [7] and characterization of properties [1]. In recent years, the research on fire behaviors of foam materials is mainly focused on polyurethane foam, with a similar structure to rubber latex foam [8,9,10,11]. The combustion of foam materials can be divided into three stages: smoldering stage, ignition to open fire stage, and fire spreading stage [12]. The smoldering stage is mainly affected by two factors: ignition energy and ignition time, and the second stage is affected by heat release from carbonization and the oxygen content of the material [13,14]. The flame spread behaviors of polyurethane foam are mainly affected by density [15], water content [15], ignition location [16,17], thickness [17], and thermal properties [15,16]. Lefebvre [15] indicated that the flame spread rate and the peak heat release rate of polyurethane foam decreased with a higher density. Robson [17] obtained that the flame spread rate increases with a bigger foam thickness. Ezinwa [16] indicated that the heat release rate of central ignition is 14% higher than that of edge ignition. Rubber latex foam and polyurethane foam are very similar in structure and application, but their different compositions may cause different fire behaviors. Therefore, these studies on fire performance of polyurethane foam may help us to address the flame spread behaviors of rubber latex foam to a certain extent.

The effect of bottom ventilation [18], ignition position [19], and layer distribution [20] on the flame spread of rubber latex foam has been studied by previous research by our group. It indicated that materials under bottom ventilation (BV), laminated samples, and center ignition position showed higher fire risk by comparing the value of flame temperature, flame height, and mass loss rate, and others. However, futher further studies should be conducted to investigate the flame spread behaviors of rubber latex foam with different thicknesses has not been studied yet. 

Rubber latex foam with different thickness is usually used as the upper filling layer of the mattress, and its bottom is directly connected with the other cushion filling layer and iron spring layer. In our experiments, the rubber latex foam was placed on the wire mesh to simulate bottom ventilation, which is the same as the actual situation. In this paper, the combustion behaviors of rubber latex foam with three different thicknesses (1, 2, and 5 cm) were compared by using a small-scale experiment platform, properties such as flame propagation speed, flame height, flame temperature, mass loss rate, and so on, to address the fire risk of rubber latex foam in practical applications. At the same times, the research results also provide a theoretical basis for fire risk assessment of typical cushion materials under spontaneous combustion conditions.

## 2. Experimental Methodology 

The small-scale platform which was developed by our group was adopted in this experiment, as shown in Figure 3a, which mainly includes the smoke hood, thermocouples, sample holder, data acquisition, chassis, electronic scale, chassis, and three CCD cameras. The experimental system, from bottom to top, includes electronic scale, chassis, stent, sample holder, electronic igniter, and smoke holder. The electronic scale (accuracy = 0.2 g, and the weighing range is 0–6 kg) is used to show the mass change of samples, which is connected with the sample holder by a stent. It should be mentioned that a chassis (0.8 m × 0.8 m × 0.005 m) is placed above the electronic scale to hold the fallen residue and avoid the influence of high temperature on the electronic scale. The sample holder was 0.3 m above the chassis, with a dimension of 0.8 m × 0.8 m. A stainless grid, made of 0.6 mm diameter stainless with a gap of 10 mm, is placed on the hollow sample holder, which is used to support the NR rubber latex foam with the same bottom ventilation. An electronic igniter fueled by liquid *n*-butane (C_4_H_10_), with a calorific value of 124 MJ/Nm^3^, was used to ignite the center of the samples with different thicknesses. In our experiments, the electronic igniter had an angle of 45° from all samples’ top surface, which makes the contact area between the electronic igniter flame and the sample surface the same. Above the sample, a smoke hood with a dimension of 2 m × 2 m × 0.8 m (height) was used to gather and exhaust the smoke during the whole experiment. A 0.4 m diameter smoke exhaust pipe was connected between the hood and a smoke exhaust fan, which can benefit the smoke exhaust process.

The measurement system includes a thermocouple data acquisition system, an electronic scale-computer acquisition system, and three CCD cameras. During each experimental test, the temperatures along the horizontal direction of the sample were recorded by six K-type 0.5 mm diameter thermocouples with a measurement range of −200 to 1300 °C. The material was ignited at the center point and, as time goes on, flame spread from this ignition point to the four edges. Hence, those thermocouples of T1–T6 were placed along a half-diagonal of the top surface from the center to the edge, and it is emphasized that all thermocouples are arranged at the upper surface of samples with different thicknesses, as seen in Figure 3b. The universal several bus (USB) interface of 34908A module data acquisition was used to connect a personal computer for easy viewing and recording the temperature. The electronic scale is placed under the chassis connecting the computer to record the real-time mass data of samples. It should be emphasized that the value shown by electronic scale is the net weight of samples. Moreover, three CCD video camera with an angle of 45°, 0°, and −45° from the horizontal were used to record the combustion processes. We draw a parallel line every 2.5 cm, with a black marker, at the surface of the specimen, to evenly divide the rubber latex foam into square pieces with sides of 2.5 cm × 2.5 cm, as seen in Figure 3a, which is not only helping us to determine the location of the fire front, but also facilitating the proportional relationship between the picture space and the actual space. Based on this, the relevant image processing software can be used to obtain the characteristic parameters of the flame, such as flame height and flame front position. The introduction of this small-scale platform is also detailed in [18].

The rubber latex foam with the thickness of 1, 2, and 5 cm were prepared. All species are cut with the small pieces of 25 cm × 25 cm. Due to the strong regularity of pore distribution of the rubber latex foam, the cutting machine is used to cut the samples into experimental materials with a uniform edge and same pore distribution according to the preset size, ensuring the size and porosity for each sample are consistent. In this study, the rubber latex foams with the thickness of 1, 2, and 5 cm were prepared, and the samples are divided into 100 small squares with black marker pen to observe the change of the position of pyrolysis, as shown in Figure 3b. All samples were selected from the same piece of rubber latex foam material with the same air pore size distribution to reduce the effect of uncontrollable parameters on the combustion characteristics of the sample. 

Samples were placed in a drying oven for 12 h before the experiment was started, under the condition that the values of temperature and humidity are 20.00 °C and 40%, respectively. This ensured that the moisture content of the samples was the same. Subsequently, they were placed on the stainless grid with holes of 10 cm × 10 cm, which enable the bottom of samples to be under the same ventilation conditions, and the surface of the sample is blown with high-speed air to remove the surface dash. In the course of the experiment, each time, we used the same lighter to ignite the center of the sample while ensuring the other conditions were the same. In order to test the accuracy of the results, each experiment was repeated twice, and showed good reproducibility, and one of the groups of experimental data among them was used for in-depth analysis.

## 3. Results and Discussion

### 3.1. Flame Behaviors

Comparison of the flame shapes of different thickness samples is helpful to explore the difference of the surface flame propagation behaviors. The variation of the rubber latex foam flame sequences with thickness *d* = 1, 2, 5 cm is shown in Figure 4. Three CCD video cameras, placed in the designated place, recorded the whole process of the fire spread, which helped us to clearly observe the three different periods of the natural rubber foam combustion: stage I is the initial combustion phase stage (from the moment that the center point of material was ignited to the fire spread across the entire top surface); stage II, the so-called “stable combustion stage”(from the end of initial combustion phases stage to the moment that flame spread to all rubber latex foam surfaces); stage III is the fire decaying stage (from the end of stage II to fray-out of flame moment), which is very similar to the combustion processes of polyurethane foam [21,22,23]. 

The combustion process of all three kinds of materials can be divided into these three stages, and the biggest difference is the duration of each stage. The duration of each stage under three thickness conditions and the average flame height is highlighted, as seen in Table 1. The three-stage combustion duration of rubber latex foam with a thickness of 5 mm is the longest (357 s), compared with the others (1, 2 cm). As is apparently shown by the pictures above, stage I, the fire began to spread from the center spot to the edges. It is worth emphasizing that the bottom of the 1 and 2 cm samples are burned out quickly while, on the contrary, the bottom of 5 cm one was not ignited. It can be seen from the flame spread images of samples with a thickness of 1, 2, and 5 cm, in the previous 62 s, that the flame spread rates of 1 and 2 cm samples are faster than that of 5 cm one. During the stage II, all samples burn steadily, and the samples burn more intensely than in the stage I, and ample with 2 cm thickness burns most fiercely, and the average flame height (0.68 m) is higher than the materials with thickness of 1 cm (0.53 m) and 5 cm (0.59 m), as shown in Table 1 and Figure 4. As the burning process progresses, the mass of samples decreases sharply in the stage III. At the end of the burning process, more residues attached to the wire mash were observed in the 5 cm materials, than the others.

There is a similar trend during the stage III that the flame gradually decreased until the fire was extinguished when the rubber latex foam was gradually consumed. Meanwhile, we can clearly see that some sample residues attached to the stainless steel mesh, and more residues were observed attached to the wire mash of 5 cm materials, than the others.

Figure 4 reveals that the bottom of the three thickness samples was ignited during the experiments, but the flame spread route is much different. In Figure 5, the red color represents the burning area, the blue one represents a no burning region, and the arrows represent the trend of flame spread. The flame of samples with a *d* = 1, 2 cm spread from the center to four edges. On the contrary, first of all, for the 5 cm sample, the fire spreads from the edge to the surrounding area. After a period of time, the bottom center is burned through, and the fire spreads from the center and edge to the middle layer at the same time. The phenomenon shows that the bottom flame spread law of rubber latex foam with different thickness is different, and the unburned zone, during the second stage, is located in four edges and the middle layer, respectively. We will further explore the parameters of flame height, mass loss rate, and temperature and flame spread rate.

### 3.2. Flame Spread Rate

The position of the flame front was obtained by image sequences as seen in Figure 6. The flame front for different thickness samples increases along the time showing a linear relationship. As we know, the rate of flame spread can be calculated via distance divided by time. Hence, it can be calculated by the reciprocal of the fitting line slope. The equation of three fitting curves obtain from this experiment are *y* = 3.43*x* + 9.06, *y* = 3.41*x* + 10.53, and *y* = 6.06*x* + 17.44, which show that the slope of 1, 2, and 5 cm test are 3.42, 3.41, and 6.06 cm/s. Correspondingly, the flame spread rate on the upper surface of the 1 cm test, 2 cm test, and 5 cm test are about 0.292 × 10^−2^ m/s, 0.293 × 10^−2^ m/s, and 0.165 × 10^−2^ m/s respectively. For the test, the flame spread rate of rubber latex foam with a thickness of 1 cm and 2 cm is similar and far greater than that of the 5 cm one, which may indicate that samples with a thickness of 1 cm and 2 cm show higher fire risk than that of 5 cm.

### 3.3. Mass Loss 

The change of sample mass is reflected by the electronic scale-computer data acquisition system. Meanwhile, the mass loss rate of samples with *d* = 1, 2, and 5 cm is the first derivative of mass versus time, which is depicted in Figure 7. Obviously, the quality of different thickness samples all decrease slowly and then decrease sharply, and the mass loss rate of samples with *d* = 2 cm is the greatest. The peak mass loss rates of samples with d = 1, 2, and 5 cm are 1.83, 1.91, 0.72 wt%/s (wt% denotes weight percentage), respectively, which all appear near the end of the second stage, and the mass loss rate increased significantly after the bottom ignited.

### 3.4. Flame Height

During the experiments, the change of flame height was reflected by CCD cameras. For the rubber latex foam with a thickness of 1, 2, and 5 cm, the flame height was measured from the moment the center point of material was ignited to flame out. The variation of flame height is shown in Figure 8 (the frequency of flame fluctuation is reflected by the standard deviation of flame height).

Figure 8 represents the change of flame height with times under different thickness. Based on Figure 4, the burning process can be detailed, and the average flame height for the sample with different thickness is 0.53, 0.68, and 0.59 m respectively, as shown in Table 1. During stage I, the thickness has little influence on the flame height. During stage II, the flame height of the sample reached its maximum for all thickness samples. Clearly, the maximum flame height of 2 cm thick sample is higher than the others. This is possible because the bottom of the 2 cm sample burns out quickly and has more combustibles than 1 cm, and the 5 cm one is not burns out immediately. Therefore, the 2 cm one shows higher flame height than that of 1 and 5 cm. Meanwhile, an obvious phenomenon deserves our attention, the stable stage duration of samples with different thickness is longer with increasing sample thickness, and the stable combustion stage of 5 cm one is much longer than that of the others. In the last stage (stage III), the value of the flame height dropped sharply with residue, and this process has the same tendency with all thicknesses (1, 2, and 5 cm) of rubber latex foam. 

### 3.5. Temperature Profiles

Flame spread is the process that the test specimens are heated and ignited by the energy from the burned region [24]. In this paper, understanding the temperature profile contributes to exploring the regulation of flame spread with different thicknesses of rubber latex foam. Figure 9 shows the temperature profiles of the rubber latex foam surface with different thickness (*d* = 1, 2, and 5 cm) during the flame spread process. As we have seen in Figure 9a–c, all the trends of surface temperature increased firstly then declined, which is similar to the flame spread regulation of polyurethane foam [23]. It can be shown in Figure 4 and Figure 9 that the flame spreading over natural rubber latex foam surface undergoes three stages. 

Figure 9 indicated that the temperature change trend and the surface average temperature are very similar during stage I for the different thickness samples. The maximum temperature at most of TC (thermocouple) point is around 400.00–500.00 °C. This is possible be because those thermocouples were located in the pyrolysis region. However, a special phenomenon was observed, that the temperature of TC6, as seen in Figure 9c, increases sharply from the lower one to 680.00 °C. This was probably because the edge of the sample shrunk, resulting in TC6 becoming separated from the surface and shifting rapidly to the flame region.

During stage II, according to Figure 9a,b, the temperature under the conditions of the thickness of 1 and 2 cm reach a peak value, which is around 700.00–900.00 °C. However, for the 5 cm one, it remains stable at about 600.00 °C, except for TC6, which reaches the maximum at an early time of the third stage, as shown in Figure 9c. There is a marked increase of change in temperature of all thermocouples under the three different conditions, which all increased from 400.00 °C to the maximum value. Moreover, the jump of the temperature of six thermocouples that are located on the top surface of samples with different thicknesses (*d* = 1, 2, and 5 cm) have been compared in the Table 2, and the average jump of the temperature of six thermocouples is 240.76, 296.22, and 237.77 °C respectively, which can show the intensity of combustion. That is to say, the 2 cm one burn more fiercely.

The reason for these differences is that the 1 cm and 2 cm materials are burned out quickly, and the top and bottom surfaces of the samples are burned simultaneously, which makes the combustion more intensely. On the contrary, because the sample with a thickness of 5 cm initially burned only on the top surface, there is a longer stable combustion stage before the bottom combustion.

Based on the bottom combustion recording by camera c, we have noticed that the bottom ignition time for the three samples is 49, 78, and 257 s, respectively. Moreover, the maximum temperature of the 2 cm sample is 948.13 °C, which is higher than that of a sample with a thickness of 1 cm (791.23 °C) and 5 cm (893.49 °C). As time went on, those thermocouples escaped from the upper surface and shifted to flame zone, leading to improvement of temperature. As the combustion came to stage III, for all thickness samples, the material was gradually exhausted, and the temperature obtained by TC1–TC6 dropped sharply.

## 4. Further Discussion

First of all, the value of thermal penetration depth (δ_t_) inflame spreading is about 3.8 cm, according to previous studies of our group [18]. When the specimen thickness is no less than the thermal penetration depth, the material is regarded as a thermally thick material. Otherwise, the result turns out contrary. Thus, in this paper, samples with a thickness of 1 and 2 cm are considered as thin-thermal material, and 5 cm thick natural rubber foam is called thermally thick material. 

The fire behaviors of thermally thin and thick rubber latex foam shows great difference, such as the law of fire spread rate at samples bottom (Figure 5), flame spread rate (Figure 6), mass loss (Figure 7), flame height (Figure 8), and temperature profile (Figure 9), which is dependent by the difference of fire spread mechanism. It has been compared, as shown in Figure 10. The mechanism of flame spread is determined by heat feedback, which includes the radiation and convective [21,25]. Samples are often classified by three parts, i.e., pyrolysis zone, preheating area (δ_f_), and virgin area. During the process of fire spread, because of the preheating effect of the flame front, the samples show a preheating zone. In the stage I, the thermally thin rubber latex foam (δ_f_) was burnt through quickly enough, which is determined by the thermal radiation, and both top and bottom surfaces begin to burn, resulting in the appearance of two preheating zones. Thermally thick materials are showing the great difference. In contrast, for thermally thick materials, the bottom of thermally thick one is not burnt through, and the flame spread was affected by both thermal radiation and heat conduction, resulting in there being only one preheating zone. Therefore, the flame spread rate of thermally thin materials is faster. 

The TG test results show that after the combustion of the rubber latex foam, the residual residue is about 24.8% of the original, which is not a thermoplastic material [3]. Moreover, due to the different combustion modes, the existence of the residues was found to be different. As shown in Figure 11, a half-section view, the yellow color represents the rubber latex foam and black represents residues after combustion, and some neat holes represent the through-holes of rubber latex foam rubber with an aperture of 6 mm and a spacing of 30 mm. The thickness of the residue formed during the combustion of the rubber latex foam of d = 5 cm is larger than that of 1 and 2cm one. As shown in Figure 11, the residue of thermally thin rubber latex foam (*d* = 1, 2 cm) is thinner. Consequently, the relative surface area of air contact with thermally thin rubber latex foam is larger than that of thermally thick rubber latex foam, so that more fresh air can help the burning, that is to say, thermally thin rubber latex foam can burn more fully than the thick one. Moreover, because of the different distribution patterns of combustion residues, the flame spreading modes of thermally thick and thin rubber latex foam are different. As shown in Figure 6 and Figure 7, the value of flame spread rate and mass loss rate of thermally thin materials is larger than the thick one. At the same time, during this burning process, the value of profile temperature and flame height of samples with *d* = 2cm are the largest, as seen in Figure 9, Table 2, Figure 8, and Table 1. In other words, the fire spread speed of thermally thick material may be lower than that of thin ones.

## 5. Conclusions

In this work, the flame spread mechanisms of rubber latex foam with different thickness (*d* = 1, 2, and 5 cm) were explored experimentally. The main conclusions include:(1)First of all, the flame spread rate of rubber latex foam with *d* = 1, 2, and 5 cm are about 0.292 × 10^−2^, 0.293 × 10^−2^, and 0.165 × 10^−2^ m/s. Moreover, the maximum value of flame height is 578.98, 852.875, and 810.264 mm. At last, the maximum temperature of the top surface is 791, 948, 893 K, respectively. The maximum mass loss rate is 1.83, 1.91, and 0.79 g/s. A variety of experimental phenomenon indicates that the NR rubber latex foam with a thickness of 2 cm shows higher fire risk, which also can be demonstrated by the value of average flame height (0.53, 0.68, and 0.59 m).(2)The bottom flame spread law of thermally thin and thick materials is different, and the unburned zone, during the stage II, is located in four edges and the middle layer, respectively.(3)Thermally thin and thick rubber latex foam produces two and one preheating zones, respectively, in the process of flame spread, which may cause high flame spread rate and high fire risk of the thermally thin rubber latex foam.(4)Due to the different combustion modes, the existence of the residue is found to be different, and because of the different distribution patterns of combustion residue, the flame spreading modes of thermally thick and thin rubber latex foam are different. The thickness of the thermally thin surface is thinner than that of the thermally thick one.

## Figures and Tables

**Figure 1 polymers-11-00088-f001:**
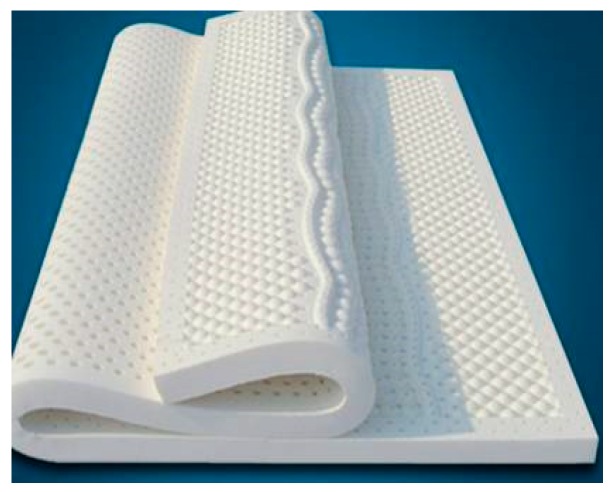
Rubber latex foam structure under naked eye observation.

**Figure 2 polymers-11-00088-f002:**
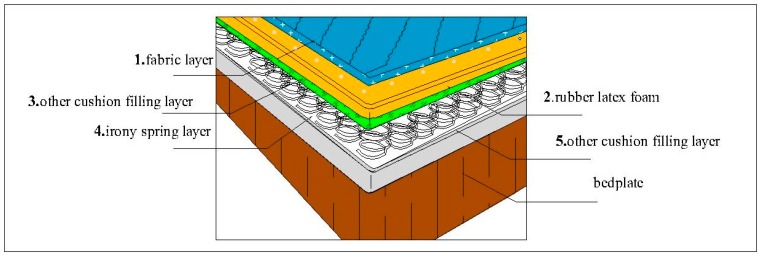
The layered structure of common rubber latex foam mattress.

**Figure 3 polymers-11-00088-f003:**
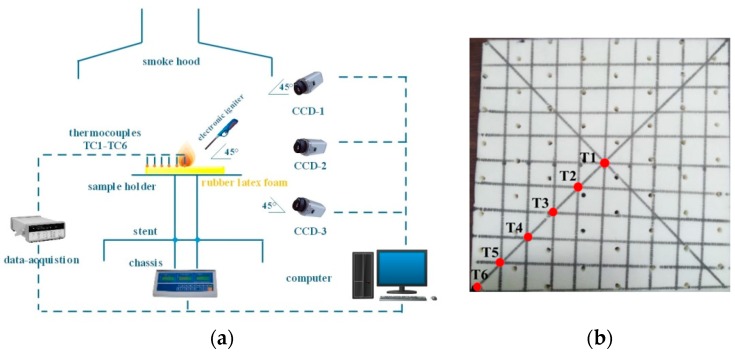
(**a**) Schematic of small-scale platform in this study; (**b**) Tested rubber latex foam sample and thermocouples positioned along the horizontal direction (T1 to T6).

**Figure 4 polymers-11-00088-f004:**
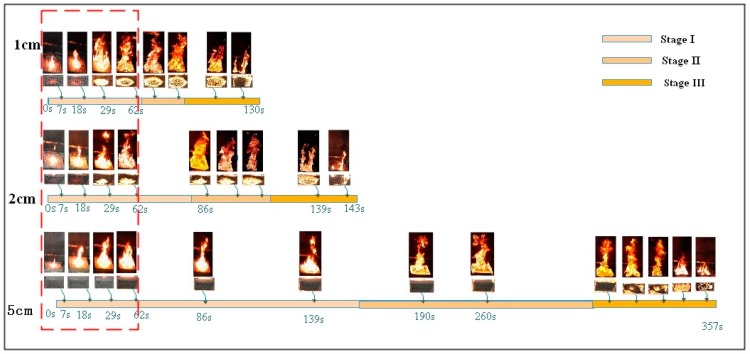
Flame spread process of rubber latex foam with different thickness (*d* = 1, 2, and 5 cm).

**Figure 5 polymers-11-00088-f005:**
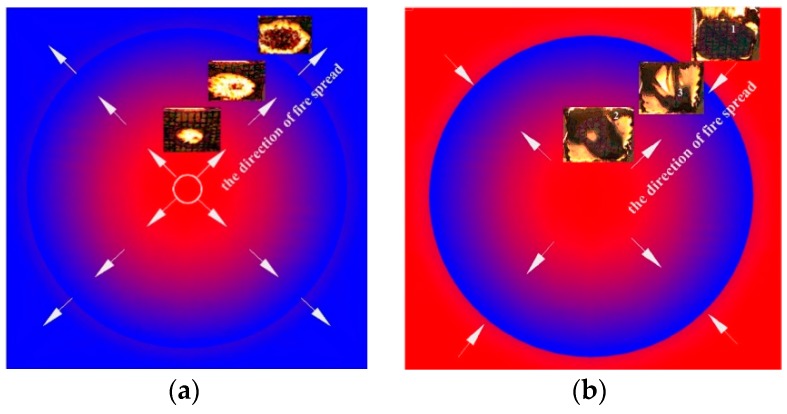
The flame spread law at the bottom of the sample with different thickness: (**a**) 1 cm and 2 cm; (**b**) 5 cm.

**Figure 6 polymers-11-00088-f006:**
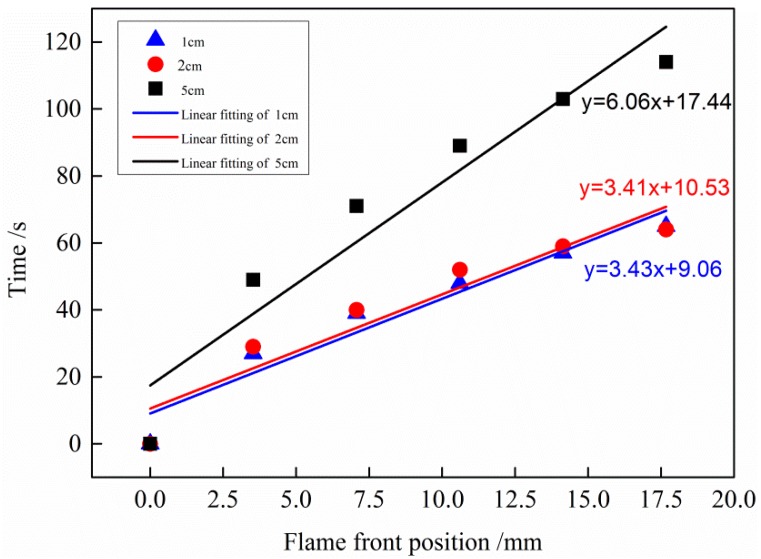
Flame front versus time under the three thickness tests.

**Figure 7 polymers-11-00088-f007:**
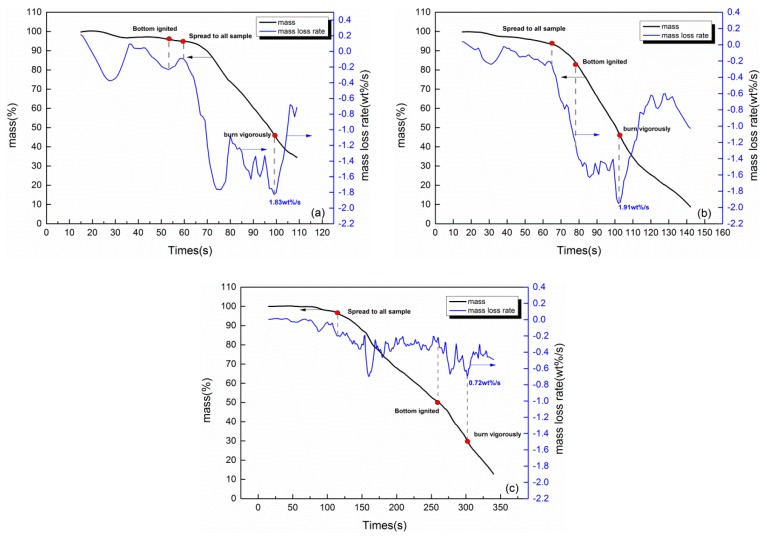
Mass loss rate of different thickness samples for: (**a**) 1 cm; (**b**) 2 cm; and (**c**) 5 cm.

**Figure 8 polymers-11-00088-f008:**
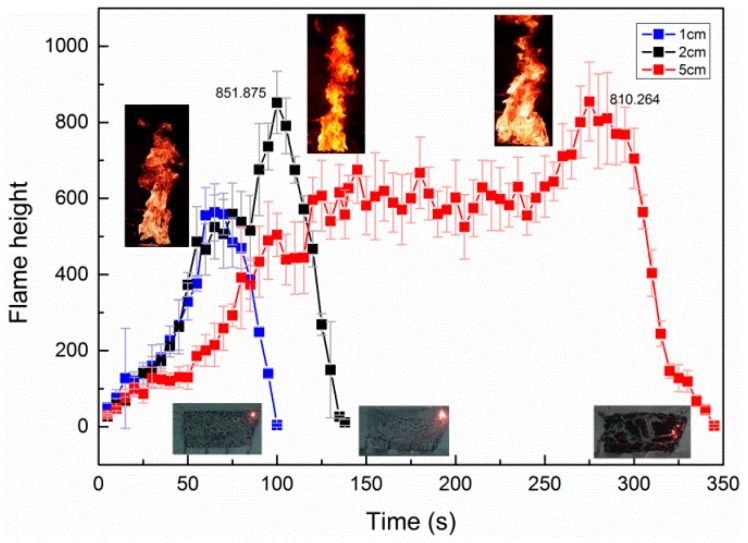
Flame height versus time for the three thickness samples.

**Figure 9 polymers-11-00088-f009:**
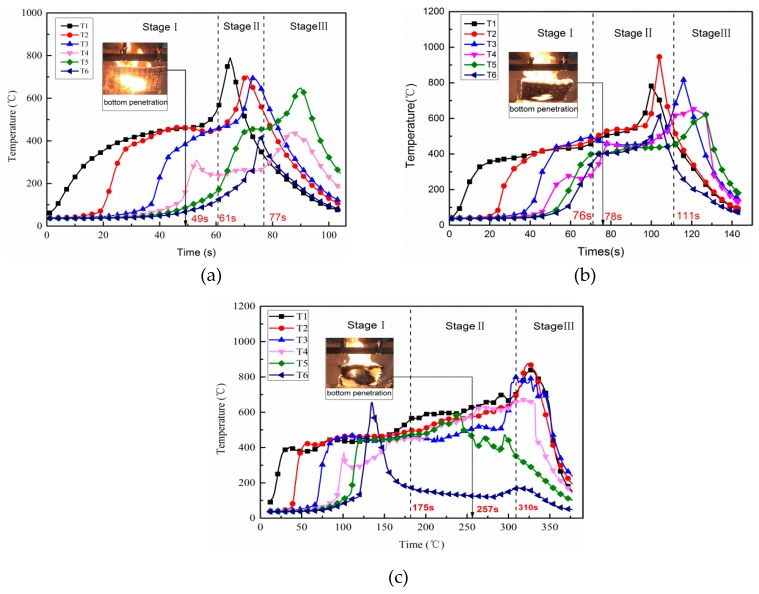
Surface temperature field changes of natural rubber latex foam: (**a**) 1cm; (**b**) 2 cm; (**c**) 5 cm.

**Figure 10 polymers-11-00088-f010:**
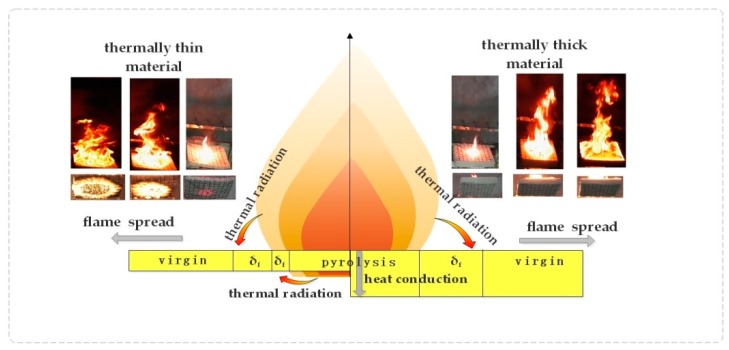
Fire spread mechanism of thermally thick and thermally thin materials.

**Figure 11 polymers-11-00088-f011:**
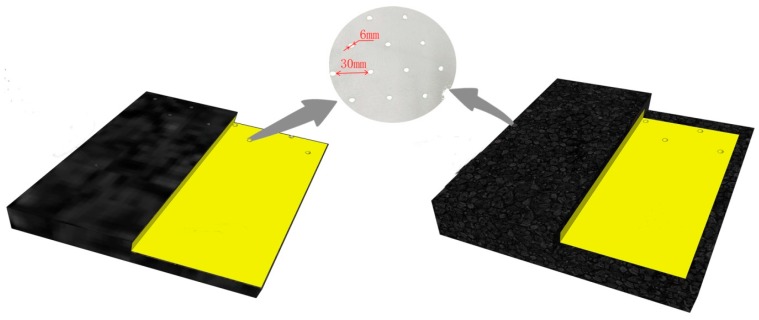
Residues distribution on the sample surface during the combustion.

**Table 1 polymers-11-00088-t001:** The duration of the three stages and the average flame height for the three thickness samples.

	1 cm		2 cm		5 cm	
	Duration time (s)	Average flame height (m)	Duration time (s)	Average flame height (m)	Duration time (s)	Average flame height (m)
Stage I	0–62	<0.53	0–75	<0.68	0–174	<0.59
Stage II	63–77	0.53	76–111	0.68	175–310	0.59
Stage III	78–130	<0.53	112–143	<0.68	311–357	<0.59

**Table 2 polymers-11-00088-t002:** Jump of the temperature for each sample.

Thickness of Samples (cm)	T1 (°C)	T2 (°C)	T3 (°C)	T4 (°C)	T5 (°C)	T6 (°C)	Average Temperature (°C)
1	323.32	262.27	231.05	180.30	194.17	253.46	240.76
2	267.23	392.23	358.16	213.08	179.06	205.55	296.22
5	177.50	208.90	255.38	176.92	120.76	487.13	237.77

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
