# Peer review of "Comparison of Fire Behaviors of Thermally Thin and Thick Rubber Latex Foam under Bottom Ventilation"

_polymers, 2019, doi:10.3390/polym11010088_

Reviewer 1 Report

The work is very interesting and makes a serious contribution to explaining the mechanism of burning thin and thick flexible polyurethane foams. The experiment was carefully planned and implemented. Significant  differences  in the burning  of thin and thick layers of flexible foams were indicated.The article is suitable for publication i journal POLYMERS.

Author Response

Thank you very much. Please feel free to let us know if you have any further question.

Reviewer 2 Report

This paper needs a substantial revision before its publication. The authors should perform a careful English language and style revision of the manuscript.

Some of the data presented contain too many digits. For instance the data presented in the abstract and in table 2. 

Introduction

The authors have included in the introduction several references that are not relevant for the topic of the paper, while they have missed several works about rubber latex foams that could be more interesting for the topic under study.

For instance the general paragraph included from line 37 until line 39 is irrelevant and could be replaced by a more focused paragraph on fire performance concerns on foams and rubber foams. 

The authors should clearly indicate that figure 1a has been taken from the article cited in reference [3].

2. Experimental methodology

The authors should describe with more detail the testing device. It is not clear which kind of heating source they have used and in which part of the sample it was placed.

The authors should mention how many experiments they have performed on each material type.

Are the thermocouples placed at the same depth for all the samples?

3. Results

The differences between stage I and II should be better explained. The authors could add some pictures of the evolution of the flame spread of the sample with 125px thickness during the stage I. It would be interesting to have images at the same times for the three cases.

The authors should clarify how many samples they have tested. Otherwise it is not clear if the differences are significant between the different thicknesses.

The authors should unify and use kelvins or centigrade for the temperature.

In general the authors should explain better the relationship between their results and the sample thickness.

Author Response

 Thank you for your comments. For detailed replies, please refer to the Word file.

Reviewer 3 Report

The paper is very practical in the content and suitable for publication after the authors revised the manuscript.

1. On line 152, the authors should describe how to measure mass loss in detail, for example, experimental conditions.

2. On line 211, the authors should recheck the unit of the maximum temperature of 2 cm is right or not.

3. On line 234, Fig. 10 was missing. The authors should recheck it again.

4. On line 229, what phenomena did the authors want to explain from Figure 9 ? It was not so clear.

Author Response

Thank you very much for your comments. For detailed replies, please refer to the Word file.

Reviewer 4 Report

The authors should add some  elements concerning the context of the use of such foams and the link of  their tests with the standard tests for targeted applications.

Please precise if the foam is produced in the same manner for each  thickness (is it a bulk foam cutted in different thicknesses ?). It  could answer some questions concerning the possible variations of  porosity, permeability, etc.

The  authors have to correct carefully the paper (the paper should be read  by a native and the spelling errors have to be corrected).

Author Response

Thank you for your comments.For detailed replies, please refer to the Word file.

Round  2

Reviewer 2 Report

The authors have significantly improved the manuscript and I, therefore consider it suitable for publication in its present form.

Reviewer 3 Report

The manuscript is suitable for publication in the journal.

Reviewer 4 Report

Dear Authors,

due to your answers and your modifications I recommend this paper for a publication in Polymers.

Sincerly yours.